# Tobacco Smoking during Pregnancy: Women’s Perception about the Usefulness of Smoking Cessation Interventions

**DOI:** 10.3390/ijerph19116595

**Published:** 2022-05-28

**Authors:** Rita Almeida, Carolina Barbosa, Bruno Pereira, Mateus Diniz, Antoni Baena, Ana Conde

**Affiliations:** 1Portucalense Institute for Human Development (INPP), Portucalense University, 4200-072 Porto, Portugal; rita_mariares_almeida@hotmail.com (R.A.); 38639@alunos.upt.pt (C.B.); 40629@alunos.upt.pt (B.P.); 40288@alunos.upt.pt (M.D.); 2eHealth Center, Faculty of Health Sciences, Universitat Oberta de Catalunya, 08018 Barcelona, Spain; abaenag@uoc.edu; 3Tobacco Control Research Group, Bellvitge Biomedical Research Institute (IDIBELL), L’Hospitalet de Llobregat, 08908 Barcelona, Spain

**Keywords:** pregnancy, smoking-habits, smoking cessation, anxiety

## Abstract

Tobacco consumption during pregnancy is a serious public health problem due to its negative effects on fetal development and on pregnant women’s health and well-being. Thus, it is of utmost importance to plan and implement smoking cessation interventions, to prevent the negative impact of this risk factor, namely on children’s health and development. This cross-sectional study aimed at exploring the perceptions and beliefs about the usefulness of smoking cessation interventions during pregnancy, in a sample of pregnant Portuguese women. The smoking use by pregnant women, as well as the risk factors associated with tobacco smoking during pregnancy, were also analyzed. The sample included 247 pregnant Portuguese women aged between 18–43-years-old (*M* = 30.30, *SD* = 5.02): 42.5% never smoked, 18.3% quit smoking before pregnancy, 19.0% quit smoking after getting pregnant and 20.2% were current smokers. The pregnant Portuguese women who smoked during pregnancy (current smokers or who quit smoking after getting pregnant) were mostly single or divorced, with lower education levels, showed a higher prevalence of clinically significant anxiety symptoms, and perceived smoking cessation interventions during pregnancy as less useful when compared to women who never smoked or quit smoking prior pregnancy. Daily or weekly smoking cessation interventions, implemented by health professionals such as doctors, nurses, or psychologists are the ones perceived as the most useful for pregnant women. These findings provide important clues for the planning of smoking cessation interventions during pregnancy, highlighting the domains that should be carefully monitored by health professionals. Specific strategies should also be used by health professionals to promote smoking cessation considering the demands of pregnancy and postpartum.

## 1. Tobacco Smoking during Pregnancy: Women’s Perception of the Usefulness of Smoking Cessation Interventions

According to the World Health Organization (WHO) [1], tobacco smoking is a public health problem with multiple risks and consequences for the general population. Tobacco consumption directly causes the death of more than 7 million people in the world, and about 1.2 million died due to passive exposure to smoke. Additionally, the WHO [2] estimates that if tobacco prevention and control policies are not effective, in 2030 about 10 million people will die per year as a result of smoking, 70.0% of them in developed countries. In Europe, 28% of the population smoke, despite the increasing knowledge about the nefarious effects of tobacco [3]. Moreover, a recent study assessing smoking habits, which involved 204 countries, suggested that for Portugal the numbers are aligned with the overall estimates, namely between 30 and 40% for men aged 15 years or over, and 20 to 29% for women in the same age group [4]. This is especially relevant considering that women enter their reproductive years during which smoking has an increasingly deleterious impact on reproduction, affecting aspects from fertility and pregnancy outcomes to fetal and child development. In fact, the gestational period is a critical and sensitive period in child development; however, worldwide, 52.9% of women who smoke daily remain with this habit during pregnancy [5]. During this period, especially in earlier stages of the pregnancy, (i.e., when the major organs form), the fetus can be very sensitive to substance-related harm caused by tobacco [6]. For instance, several tobacco chemicals and nicotine can cross the placenta [7,8] increasing the risk of early birth by 5.0% to 8.0% [9,10,11]; limited fetal and uterine growth, resulting in lower birth height and weight in 13.0% to 19.0% of cases [10]; sudden infant death syndrome with a probability of 23.0% to 34.0% [10]; hypoxia, respiratory and neuronal diseases, among others [12,13]. Although the consequences of smoking habits during pregnancy are already well established [14] and the gestational period can be considered as a motivating factor for smoking cessation [10], the prevalence of female smokers is still relatively common and increasing mainly in young adults [15]. This has led to claims that women are in epidemic stage 2 [16] regarding tobacco smoking, which shows that the number of female smokers is still on a rise [17], especially during the reproductive years. In Spain, for example, half of the women who smoke before pregnancy continue smoking during this period, hiding this habit; it is also estimated that 25.0% of women of reproductive age are active smokers and that 70.0% of women who quit smoking during pregnancy return to this habit after giving birth [18]. A study conducted by Alves et al. [19], involving 5420 Portuguese pregnant women, showed a prevalence of 23.0% of women smokers, and more than half of the women who smoked at the beginning of pregnancy have maintained tobacco consumption during pregnancy (59.7%). Of the 48.0% who stopped smoking, 32.0% relapsed after delivery. Considering tobacco consumption, 42.0% of the participants stated to have reduced smoking from 15 to 7 cigarettes per day, and from those, only 8.0% maintained this reduction after pregnancy, with 34.0% returning to the frequency of consumption prior to pregnancy [19]. Moreover, another recent study pointed out that previous tobacco consumption can affect women’s tobacco consumption during pregnancy and the postpartum period [20], suggesting the importance of early assessment of tobacco use among women of reproductive age, as well as its determinant factors. Understanding what characterizes women who continue smoking during pregnancy would clearly lead to great potential health gains for both mother and child, and society. Namely, the knowledge could be used to tailor preconception prevention strategies as well as interventions during pregnancy focused on these risk factors and directed to high-risk pregnant women.

These risk factors include high nicotine addiction, easy access to the substance, difficulties in the access to smoking cessation interventions, and the number of cigarettes smoked per day [21,22,23]. Lower socioeconomic levels, low education, and occupational status, (e.g., unemployment, night work hours) were also identified as risk factors for tobacco consumption [13,15,22,23,24,25]. Multiparity, poor antenatal care, being a young or single mother, and unexpected and unwanted pregnancies are also pointed as associated with smoking habits during pregnancy [15,26,27]. Finally, there are also contextual factors that seem to be associated with smoking, such as exposure to environmental tobacco smoke, living without a partner or the partner being a smoker, exposure to physical or sexual violence, high levels of stress associated with the experience of stressful personal events, low social support, alcohol, or other substances consumption, as well as the presence of psychiatric comorbidity [23,26,28,29,30,31,32].

Moreover, nicotine dependence is classified in DSM-5 and ICD-10 as a mental disorder [33], with a high comorbidity with other anxiety disorders [33,34,35]. Even in low-risk samples, increases in pregnant women’s anxious symptomatology have been observed [36,37]. However, the trajectory of changes observed in anxiety symptoms from the first to the third trimester seems to be different between pregnant women who never smoked, former smokers, and who continued smoking during pregnancy. In fact, contrary to the other two groups, pregnant smokers did not show a decrease in anxiety symptoms over pregnancy [38].

As previously pointed out, the smoking habits tend to return to the pre-pregnancy levels during the postpartum period, namely affecting breastfeeding by reducing milk production, as well as changing the nutritional composition and taste of the milk. When the woman smokes at this stage, cigarette substances such as aluminum, lead, carbon monoxide and dioxide, and, mainly, nicotine are secreted into breast milk and have the potential to cause severe adverse effects on the newborn [39].

The high prevalence of tobacco consumption by pregnant women, the high rate of smoking habits relapse after childbirth (for mothers who discontinued smoking during pregnancy), and the adverse impact of smoking on mothers’ and children’s health and well-being, easily justify the need for screening of smoking habits, as well as the development and implementation of smoking cessation intervention programs in the prenatal and postnatal periods. Promoting smoking cessation in pregnant women provides an opportunity to promote women’s healthy habits and to reduce the harmful effects of tobacco exposure on child development. In fact, although reducing tobacco smoking is important to decrease its detrimental effects on mothers’ and children’s health and well-being, empirical evidence shows that the consequences of tobacco smoking only disappear when smoking is completely stopped [40]. Therefore, it is extremely important to access and monitor the consumption rates of tobacco in pregnant women, in order to inform women and demystify the belief that lower doses of nicotine will decrease the likelihood of tobacco harm [41]. Another future benefit of smoking cessation during pregnancy, even in later stages of gestation, is to reduce the likelihood (almost 50%) of children starting to smoke, which may ultimately interrupt the inter-generational cycle of smoking habits [42].

Despite the importance and benefits of smoking cessation, evidence, (e.g., [43,44]) suggests that this topic is not always properly addressed by healthcare professionals. Barriers to this include the lack of training in smoking cessation support, the lack of time to address this condition in obstetrical consultations, and the lower perception of risk associated with smoking habits by pregnant women, especially when healthcare professionals are, themselves, smokers [22,45]. Moreover, it is important to note, based on the findings of Okoli et al. [43] that, although more than 50% of the health care providers are likely to assess the smoking habits of pregnant women and advise smoking cessation, less than 50% also assess the readiness to change, or provide assistance in smoking cessation or arrange for follow up appointments or referrals. Thus, the role of healthcare professionals in planning and implementing smoking cessation interventions is of utmost importance, especially considering the findings pointing out that individuals who try to quit smoking without any type of support relapse in the first weeks after the attempt [46]. Moreover, the success of counseling techniques for smoking cessation are dependent on the motivation of the patient who smokes, her/his adherence to treatment, as well as on the quality of the monitoring process, and the therapeutic relationship [47]. Thus, it is of extreme importance to explore the perceptions and beliefs of pregnant women about the usefulness of smoking cessation interventions during pregnancy.

This cross-sectional study in nature aimed:(1)To analyze the prevalence of smoking in a sample of Portuguese pregnant women, describing their smoking habits;(2)To identify the risk factors of smoking during pregnancy, comparing pregnant smokers vs. non-smokers regarding sociodemographic, obstetric, and psychological characteristics, as well as exposure to environmental tobacco smoke;(3)To explore pregnant women’s perceptions about the usefulness of smoking cessation interventions (types of treatment methods and frequency) during pregnancy.

Considering the maternal and child risks associated with smoking, a decrease in tobacco smoking during pregnancy, compared to the pre-pregnancy period, is expected [43]. Higher levels of clinically significant anxiety symptoms and other risk factors (such as the risk of pregnancy, higher tobacco consumption prior to getting pregnant, being single/divorced, environmental tobacco smoke exposure) are expected in pregnant women who smoke or stopped smoking after getting pregnant compared to non-smokers or women who stopped smoking before getting pregnant. Regarding pregnant women’s perceptions, it is expected that most pregnant women, smokers or non-smokers, consider the implementation of smoking cessation interventions by healthcare professionals to be important.

## 2. Materials and Methods

### 2.1. Participants

A total of 247 pregnant Portuguese women (community sample) living in Portugal (both in the mainland and Madeira and Azores archipelagos), aged between 18 and 43 years old (*M* = 30.30; *SD* = 5.02) were recruited. Most of them were married, had higher education, and were, at the time of the study, employed but not working, (e.g., sick/medical leave). Regarding the obstetric information, most of the participants were primigravid, were in the 2nd pregnancy trimester (between 14 and 26 weeks), and had non-risk pregnancies (self-reported information considering what was defined in the medical records). Most of the pregnant women (*n* = 227; 91.9%) did not have a reported psychopathological disorder diagnosis. Considering the participants who reported having a disorder (*n* = 20; 8.1%), the most frequently mentioned were anxiety or mood disorders. Moreover, considering the cutoff point ≥ 40 for STAI, 58.7% of pregnant women presented high anxiety (see Table 1).

### 2.2. Procedures

This study implied a cross-sectional design. Approval by the Health Ethics Committee of the Portucalense University (CES-UTP 003) was obtained before data collection. Recruitment was implemented through a non-probabilistic snowball sampling, considering the following inclusion criteria (a) being over 16 years old, (b) being pregnant, and (c) being Portuguese. Data collection was carried out during the first COVID-19 pandemic lockdown, between May and July 2020. Data collection was performed in a single moment through an ad-hoc online survey made available in the Portuguese language through the LimeSurvey© platform. Complying with the ethical requirements established for research with human beings, participation in this study was voluntary After presenting the aims and procedures of the study, participants who agree to participate signed the informed consent, although the participant could withdraw at any time without any negative implications. Likewise, no risks were foreseen for the participant arising from participation in the study, and no incentive was given for completing the survey. The study was disseminated through personal social networks and professional scientific associations. The survey included questions concerning sociodemographic information, obstetric information, anxiety symptoms (the Portuguese version of the Spielberger State Anxiety Inventory—STAI-Y-I), smoking habits, nicotine dependence (Heaviness Smoking Index—HSI), exposure to environmental tobacco smoke, and pregnant women’s perceptions about the smoking cessation interventions during pregnancy.

### 2.3. Instruments

#### 2.3.1. Sociodemographic and Obstetric Questionnaire

The sociodemographic and obstetric questionnaire was designed for this study with the aim to collect sociodemographic data (age, marital status, location of residence, education, and occupational status), as well as participants’ psychological (diagnosis of psychological disorders and their identification) and obstetric information (such as number of pregnancies, pregnancy weeks at the time of the participation at the study, type of pregnancy (risk vs. non-risk), and gravidity).

#### 2.3.2. State-Trait Anxiety Inventory—Form Y-1 (STAY-Y)

Anxiety symptoms were assessed through the State Anxiety Inventory—Y-I form (STAI-Y) [48] adapted and validated for the Portuguese population [49]. The state anxiety is defined as a transient emotional state that varies in intensity, changing over time [49]. Thus, only the state anxiety subscale of the State-Trait Anxiety Inventory, composed of 20 self-report items, was used to assess how the participants feels at the moment of the scale’s fulfillment.

Each statement is rated by the participant using a 4-point Likert-type scale ranging from 1 (not at all) to 4 (very much). The total score of this sub-scale is obtained through the sum of 20 items, with a minimum score of 20 points and a maximum score of 80. Ten of these items (1, 2, 5, 7, 9, 11, 12, 15, 19, and 20) should be reversed before calculating the total score. The cut-off point of 40, often used to define clinically significant levels of anxiety [50], was used in this study to identify pregnant women with high anxiety symptoms vs. low anxiety symptoms.

The internal consistency of the state anxiety and trait-anxiety scales was measured by calculating the alpha coefficients and the corrected item-total correlations. The value of the state anxiety scale was 0.86 according to the Kuder–Richardson formula modified by Cronbach (1951), which proved to be satisfactory [49]. In the present study, some items were not scored by the participants. When a maximum of two values were missing, these missing values were replaced by the total mean of the items answered, multiplied by 20, and rounded to the next higher value, as suggested by Spielberger et al. [48,51]. The internal consistency of the state anxiety scale was also calculated for this study (α = 0.95).

#### 2.3.3. Heaviness Smoking Index (HSI)

The Heaviness Smoking Index (HSI) [52] assesses nicotine dependence by combining two items taken from the Fagerstrom Test for Nicotine Dependence (FTND)—“On average, how many cigarettes do you currently smoke per day?” (answer options: less than 11 cigarettes; between 11–20; between 21–30; more than 30 cigarettes) and “How long, after you wake up, do you smoke your first cigarette?” (answer options: less than 5 min; between 5–30 min; between 30–60 min; more than 1 h) [52]. Studies have shown that the HSI is equally or more effective in assessing its construct than the FTND [53,54,55] and is particularly appropriate for epidemiological investigations [56].

The score in HSI is performed through the sum of the answers in the two items, with a minimum of 0 and a maximum of 6 values. Results between 0 and 2 points are classified as low dependence, between 3 and 4 points are classified as moderate dependence, and between 5 and 6 points are classified as high nicotine dependence.

#### 2.3.4. Questionnaire for Smoking Habits and Pregnant Women’s Perceptions about Smoking Cessation Interventions during Pregnancy

A set of purpose-built questions, organized in different sections, was designed to explore smoking before and during pregnancy, assessment of smoking habits of pregnant women in routine antenatal care, and pregnant women’s perceptions about the usefulness of smoking cessation interventions during pregnancy.

To explore smoking habits of pregnant women, the questions included: the use of tobacco by pregnant women during pregnancy (smoker; quitted smoking before getting pregnant; quitted smoking after getting pregnancy; non-smoker); age of the onset of tobacco smoking (for women who smoked or who quit smoking either before or after getting pregnant); average number of cigarettes per day either before and after get pregnant (answer options: 10 or less, between 11–20, between 21–30, ≥31 cigarettes); types of tobacco products (answer options: cigarettes, smoked e-cigarettes, smoked heated tobacco (water pipes) and smoked roll-your-own cigarettes (RYO)); attempts to stop smoking (number of attempts and methods used for smoking cessation (yes/no) considering the following listed methods: without help, self-help guides and other materials; counseling for smoking cessation by a health professional, psychological treatment, combined pharmacological and psychological treatment, nicotine replacement therapy, pharmacological treatment (bupropion or varenicline), and personalized online counseling for smoking cessation). 

The set of purpose-built questions designed to explore the environmental tobacco smoke exposure included the frequency of general level of exposure (answer options: once a month or never, weekly, monthly, and almost daily), as well as exposure to tobacco consumption of other family members and visitors inside the family home. 

The analysis of the assessment of smoking habits of pregnant women in routine antenatal care questioned if pregnant women were asked about tobacco use during obstetric consultations (yes or no options) and whether any smoking cessation intervention was recommended by healthcare professionals (yes or no option selection for the listed methods for smoking cessation provided—self-help guides and other materials; counseling for smoking cessation by a health professional; psychological treatment; combined pharmacological and psychological treatment; nicotine replacement therapy; pharmacological treatment (bupropion or varenicline); and personalized online counseling). 

The last section of the questionnaire assesses pregnant women’s perceptions about which intervention(s) (using the same answer options and list of methods for smoking cessation as indicated before) and respective regularity (answer options: once a month or never, weekly, monthly, and almost daily) are considered as the most effective for smoking cessation. 

### 2.4. Procedures and Data Analysis

Statistical analyses were implemented using the software IBM SPSS Statistic 26.

To accomplish the first aim, related to the analysis of the prevalence of smoking in a sample of Portuguese pregnant women, describing their smoking habits, descriptive and frequency analyses were performed. At this level, considering the participants who smoked at some point in their lives, the age of onset of tobacco use, the number of cigarettes smoked before, after becoming pregnant, and at the time of the participation in the study (if applicable), the number of attempts to stop smoking and the methods used for this purpose were also analyzed using descriptive statistics. Additionally, for pregnant women who smoked at the time they participated in the study, the types of smoked tobacco products and the index of nicotine dependence were described using frequency analysis.

To identify the risk factors of smoking during pregnancy, comparing pregnant smokers vs. non-smokers regarding sociodemographic, obstetric (primigravid vs. multigravid; risk pregnancy vs. no risk pregnancy), and psychological characteristics (High vs. low anxiety, based on the cut-off point of 40 for STAI-S scores), as well as exposure to environmental tobacco smoke (yes vs. no), chi-square association tests were applied. Before these tests were implemented some variables needed to be recategorized. The answer options “Married”, “Single” and “Divorced” of marital status variable were recategorized into two groups: “Single + Divorced” vs. “Married”. In the education variable, the category “Primary education” was merged with the category “High school” when comparisons with “Higher education” were made. Concerning the occupational status, the categories unemployed and employed but not working were merged and compared with the category employed. The continuous variable age was transformed into a categorical variable, comparing pregnant women between 18–35 vs. ≥36 years old pregnant women. Regarding tobacco consumption, two groups were considered: non-smokers (comprising the women who answered, “Never smoked + Quitted smoking before getting pregnant”) and smokers (comprising the pregnant women who answered, “Currently smoking + Quitted smoking after getting pregnant”). 

Finally, to explore the pregnant women’s perceptions about the use and usefulness of smoking cessation treatment during pregnancy (third aim), frequency analyses were performed on whether these women consider important the development and implementation of these interventions during pregnancy, and which methods and respective regularity perceived as the most effective for them. Regarding the methods, several alternatives were given, and the participant should select yes or no to the different types of smoking cessation interventions listed before. The percentages of the agreement for each of the treatment’s alternatives are provided. Chi-square association tests were also performed to explore the associations between tobacco consumption and the perception of usefulness and the most effective frequency of the smoking cessation interventions during pregnancy. 

*p* values of <0.05 deemed significant. *p* values of <0.10 were used to report marginally significant differences.

## 3. Results

### 3.1. Description of the Participants’ Smoking Habits

When the prevalence of tobacco smoking was analyzed, 50 pregnant women (20.2%) smoked tobacco during pregnancy, 47 pregnant women (19.0%) reported having quit smoking after getting pregnant, 45 (18.3%) reported having quit smoking before pregnancy, and 105 (42.5%) answered that they had never smoked. The ages of onset of tobacco use ranged between 10 and 26 years old (*M* = 16.50, *SD* = 2.74).

Figure 1 describes the smoking habits of women before getting pregnant. As can be seen in the figure, most of the pregnant women who smoked at some point in their lives indicated that they consumed 10 or fewer cigarettes, on average, per day, before getting pregnant. However, the greater the number of cigarettes consumed before pregnancy, the greater the probability of maintaining this consumption during pregnancy. 

Focusing on women who quit smoking after getting pregnant, a reduction in tobacco consumption was observed: from 42/47 pregnant women who have answered the question, “How many cigarettes smoked on average, per day, after getting pregnant”, 40 women indicated that they smoked 10 or fewer cigarettes and 2 women indicated that they smoked 11–20 cigarettes. 

Of the participants who smoked at the time of completing the questionnaire (*n* = 50), most of them (*n* = 39; 78.0%) smoked cigarettes; two pregnant women (4.0%) smoked e-cigarettes, seven (14.0%) smoked heated tobacco (water pipes) and two (4.0%) smoked roll-your-own cigarettes. Although 39 pregnant women indicated that have attempted to quit smoking, 22.0% of women never tried to leave this habit. At the time of participation in the study, 42 (84.0%) pregnant women indicated that they smoked 10 or fewer cigarettes on average, per day, seven (14.0%) between 11–20 cigarettes, and only one (2.0%) between 21–30 cigarettes. The level of nicotine dependence of these women was also assessed through the HSI, and most showed low dependence (min = 0; max = 4; M = 1.24, SD = 1.13). Specifically, 43 (86.0%) of the participants who smoked revealed low dependence (between 0 and 2 points), and 7 (14.0%) revealed moderate dependence (between 2 and 4 points).

Considering the overall sample, most of the pregnant women (59.5%) indicated that they are exposed to environmental tobacco smoke less than once a month (or never) in general. If at home, considering smoking in the household this percentage drops to 56.5%, however it increases if the smoking habits of others who visit the family home are included (71.3%). Even though it is important to highlight the high percentage of pregnant women that are exposed daily to environmental tobacco smoke in the different contexts (20.6%), especially at home (39.8%).

### 3.2. Differences between Smokers vs. Non-Smokers Pregnant Women Considering Sociodemographic, Obstetric, Psychological and Environmental Tobacco Smoke Exposure Dimensions

Associations between tobacco smoking and sociodemographic (participant’s age, marital status, education, and occupational status), obstetric (gravidity, and pregnancy risk status), psychological (anxiety symptoms), and environmental tobacco smoke exposure dimensions were explored using chi-squared tests (see Table 2).

As presented in Table 2, significant associations were found between tobacco smoking habits of the pregnant women and marital status, χ^2^ (1) = 4.42, *p* < 0.05; prevalence of clinically significant anxiety symptoms, χ^2^ (1) = 4.54, *p* < 0.05; and exposure to environmental tobacco smoke, χ^2^ (3) = 13.28, *p* < 0.01. Marginally significant associations were also found between tobacco consumption habits of pregnant women and education, χ^2^ (1) = 3.02, *p* < 0.10. Pregnant women who smoked (either during pregnancy or that quit smoking after getting pregnant) were mostly single or separated/divorced, lower educated (with primary education or high school), and with a higher prevalence of anxiety at clinically significant levels than the non-smoking pregnant women (this is, the pregnant women who had never smoked or quit smoking before getting pregnant). Moreover, the results also indicated that pregnant women who were smokers are more likely to be exposed to environmental tobacco smoke in general than the non-smoking pregnant women (see Table 2).

### 3.3. Tobacco Smoking Assessment in Obstetric Consultations and Pregnant Women’s Perceptions about the Usefulness of Smoking Cessation Interventions during Pregnancy

When an assessment of tobacco consumption during regular obstetric consultations was explored, 217 (87.9%) pregnant women indicated that they were asked about smoking when they went to the obstetric consultations, a higher number compared to the 30 (12.1%) pregnant women who indicated that their tobacco consumption habits were not assessed during the regular obstetric consultations. No significant associations were found between the responses in this question and smoking in pregnant women, χ^2^ (1) = 0.51, *p* = 0.48. Moreover, focusing on pregnant women who smoked during pregnancy (*n* = 97), most of them indicated that no recommendation was given for smoking cessation treatment by the health professional during the regular obstetric care consultations. Of the 16 women to whom this counseling was given (16.5%), the method for smoking cessation more frequently referred (by 11 smokers) was counseling for smoking cessation implemented by a health professional. Recommendation of combined pharmacological and psychological treatment (*n* = 2), self-help smoking cessation materials (*n* = 4), psychological treatment (*n* = 2), nicotine replacement therapy (*n* = 3) and pharmacological treatment (*n* = 1) were also selected by pregnant women who smoked.

Regarding the pregnant women’s perceptions about the usefulness of smoking cessation interventions during pregnancy, and considering the overall sample, 85.8% of participants considered that the development of smoking cessation programs during pregnancy is important. Differences were found between smokers and non-smokers in terms of their perceptions about the usefulness of the development of smoking cessation interventions during pregnancy, χ^2^ (1) = 5.36, *p* < 0.05. The importance attributed to the development of smoking cessation interventions was more signaled by those who had never smoked or stopped smoking before pregnancy (89.9%), than by pregnant women who smoked during gestation or quit smoking after becoming pregnant (79.4%). Moreover, weekly (59.8%) or daily (29.1%) interventions were perceived by pregnant women as the most useful and effective for smoking cessation, although some pregnant women have not answered these questions (*n* = 1 and *n* = 58 missing values, respectively). No significant differences were found between smokers and non-smoking pregnant women at this level, χ^2^ (4) = 5.09, *p* = 0.28.

Participants who considered that smoking cessation interventions during pregnancy were important (*n* = 211), regardless of being smokers or non-smokers, were questioned about which methods they considered to be the most useful. As can be seen in Figure 2, the implementation of smoking cessation counseling by a health professional, (e.g., doctor, nurse, psychologist) was perceived as the most useful and effective smoking cessation intervention for a high number of pregnant women (141; 66.8%), 90 (67.2%) non-smokers and 65 (66.2%) smokers).

## 4. Discussion

This cross-sectional study aimed at describing the smoking habits of a sample of Portuguese pregnant women, as well as identify risk factors related to tobacco consumption during pregnancy. Another important aim was to explore the tobacco consumption screening by health professionals during the routine obstetric consultations, as well as pregnant women’s perceptions about the usefulness of smoking cessation interventions during pregnancy. Perceptions about the most effective methods and regularity of smoking cessation interventions were also explored.

As initially hypothesized and in line with previous findings [20,48], a decrease in tobacco consumption during pregnancy, compared to the pre-pregnancy period was found. Higher prevalence of clinically significant anxiety symptoms and other risk factors (such as being single/divorced, lower education levels, and higher environmental tobacco smoke exposure) were found in pregnant women who smoke or stopped smoking after getting pregnant compared to non-smokers or women who stopped smoking before getting pregnant. Contrary to the previous literature, neither age nor occupational status or pregnancy risk was associated with smoking habits. This could be related to the specific characteristics of the pregnant women who constituted the sample, mainly with non-risk pregnancies (considering the criteria of age and obstetric risk). Moreover, the criteria used to define the groups for occupational status, adding unemployed and employed, but not working pregnant women in the same group, have attenuated the psychosocial risk associated with unemployed pregnant women. Regarding the pregnant women’s perceptions about the usefulness of smoking cessation interventions during pregnancy, contrary to what was expected, non-pregnant women perceived this intervention as more useful than the pregnant women who did not smoke during gestation. However, it is important to highlight that this latter group may have considered the smoking cessation interventions less useful, by the simple fact that they had already stopped smoking.

Tobacco consumption is considered a public health problem due to its high prevalence among consumers and its association with health problems [3]. In Portugal, the age group of 25–34 years old, overlapping with the reproductive years and higher fertility, is the one with the highest prevalence of tobacco consumption [57]. In this sense, it is important to acknowledge that this behavior can lead to harmful effects during the gestational period, which can in turn lead to serious consequences both for the pregnant women and their offspring. 

Despite the well-known deleterious and severe effects of tobacco consumption during pregnancy [9,10,11,12,13], the results of this study suggested a high prevalence of smoking habits among pregnant women (20.5%). Nevertheless, this cannot be disentangled from a recent event that caused a major change in the way people establish their routines, namely the COVID-19 pandemic. For instance, increases in tobacco consumption in daily smokers during COVID-19 lockdown were associated with psychological distress and anxiety [58]. As such, this high prevalence of tobacco consumption by pregnant women can also be explained, at least partially, by restrictions and routine changes imposed during the lockdown periods. Still, the level of nicotine dependence was rather low, with women smoking 10 or fewer cigarettes per day. Despite the reduction in tobacco consumption during pregnancy, in line with previous findings [59], this does not seem, however, to mitigate the adverse effects of tobacco consumption, even at low doses [34]. 

Several risk factors were associated with tobacco consumption during pregnancy. In line with the literature, in this study, pregnant women who smoke were mostly single [26,29] and with lower education levels (primary education and high school) [13,15,19,23,60]. It can be hypothesized that the lower educated women are less informed about the harmful consequences of tobacco consumption, as well as of the diversity of smoking cessation interventions available to support them in the adoption of healthy behaviors during the prenatal period. The lack of health literacy can also impose on these women higher difficulties in accessing the health care system, putting them (mothers and children) at higher risk of health problems. 

Regarding anxiety symptoms, it was found that pregnant women who smoked during pregnancy had a higher prevalence of clinically significant anxiety symptoms than women who never smoked or quit smoking before getting pregnant. This is in line with previous findings suggesting that anxiety is associated with women’s tobacco smoking during pregnancy and the postpartum period [20,61]. It has also been concluded that dependence and withdrawal syndromes play a key role in the stress/anxiety-tobacco consumption relationship [62]. In fact, although nicotine can act as a relaxant agent reducing anxiety levels [35], this relaxing effect is rather short-lived and then the organism starts to activate and go on this anxiety-related frenzy. This cycle is perpetuated by a negative reinforcement mechanism, leading to an increase in tobacco use to reduce anxiety and to an increment in anxiety levels in the face of tobacco withdrawal. Additionally, the high comorbidity between substance consumption, such as tobacco, with other psychological disorders like anxiety and depression disorders [63] should be highlighted, which could add up as increased risks, especially in a transition period marked by an increase in anxiety, such as pregnancy [40]. These findings support the need of assessing anxiety and using anxiety-specific interventions during smoking cessation interventions [41]. Furthermore, future studies should explore the direction of the causal effects between anxiety and tobacco consumption. 

Pregnant women who are smokers also exhibited a great exposure to environmental smoking than the ones that are non-smokers, which is again in line with previous studies [23]. One of the statements in Article 14 of the WHO Framework Convention on Tobacco Control (WHO FCTC) is, precisely, that women have the right to a smoke-free environment, at home, workplace, and in public places [8]. At this level, the present results highlight the non-compliance with this regulation and the need to protect pregnant women, to prevent the consequences of tobacco consumption both for their health, as well as for their children’s health and healthy development.

To guide the efficient design and implementation of smoking cessation interventions during the gestational period, the pregnant smoker women’s use, as well as the perceptions about the usefulness of different types of smoking cessation interventions during pregnancy was explored. At this level, it is important to highlight that, although pregnant women were asked about substance consumption (such as tobacco) during the regular obstetric follow-up consultations, most of them indicated not having received any recommendation for smoking cessation interventions. In line with previous findings, it is possible that these findings are related to health professionals’ attitudes and misconceptions that undermine the implementation of smoking cessation interventions, such as attitudes toward existing smoking cessation interventions, acceptance of the patients’ smoking use, and perceiving the patients’ smoking as a useful tool, namely for decreasing anxiety [43,44]. Is also possible that pregnant women who smoked during pregnancy did not accurately recall whether they received referrals or simply that they chose to ignore the referrals because they were not motivated enough to stop their consumption habits. This hypothesis is in line with the findings of a study conducted by Griffiths et al. [64], in which pregnant women from the United Kingdom reported reduced knowledge of the available services for smoking cessation and suggested several barriers such as environmental context and resources, social influences; pregnant women beliefs about their skills to reduce their smoking habits, and about the consequences, intentions, and emotions associated to smoking. 

Whatever the reason, the results found in the present study are worrying and sustain the increased risk of health and developmental disorders affecting pregnant women who smoke during pregnancy, as well for their children, who will not be a target for medical and psychological interventions, especially in a critical and sensitive developmental period [65]. Thus, early screening of tobacco consumption and the referral of pregnant women who smoke during pregnancy to specialized smoking cessation interventions creates an excellent window of opportunity for intervention, especially when women are more motivated to adopt healthy behaviors. However, previous studies suggest some barriers that are frequently mentioned by the healthcare professionals to justify this non-referral, such as the lack of training in smoking cessation interventions [66], the lack of time to address this issue in obstetric consultations, and the possible low perception by the healthcare professionals that pregnant women have of the risk associated with tobacco consumption, especially when healthcare professionals are, themselves, smokers [22,45]. Another result that should be given special attention, due to its potential increased associated risk, is the one showing that the pregnant women who smoke are those who perceived the smoking cessation interventions during pregnancy as less useful compared with the pregnant women who are non-smokers. It is hypothesized that this result could be related to a defense mechanism related to cognitive dissonance. In other words, pregnant women who are smokers will perceive the involvement in smoking cessation interventions as less useful in order to be consistent with their behavior; and by doing that, they will not have to be aware of the dangers and harmful effects of smoking by health professionals, and therefore they don’t have to quit smoking [67]. 

Results also suggested, based on participants’ perceptions, that smoking cessation interventions would be more effective during pregnancy if implemented by a healthcare professional (doctor, nurse, psychologist…) and on a daily or weekly regularity. These results should be further explored in future studies, either in Portugal or in other countries, to establish evidence-based interventions for successful smoking cessation during the perinatal period.

Supported by previous findings pointing out that about 60–80% of women who quit smoking during pregnancy returned to the smoking habits within 1 year after delivery [68], the present results also sustain the importance of considering in future studies women’s perceptions about several aspects of smoking cessation interventions, such as the type of smoking cessation treatment to be used, its regularity, women’s motivation to participate in these interventions, determinant factors of treatment adherence during pregnancy and postpartum periods, benefits of the implementation of a postpartum follow-up, (e.g., baby consultations), and so on in order to reduce the high percentages of postpartum relapse. In fact, developing effective smoking cessation interventions during pregnancy seems also have future benefits for the offspring, considering the reduced likelihood (almost 50%) of children starting smoking if the mother is not a smoker, which may ultimately interrupt the cycle of smoking from generation to generation [42]. 

Although the important contributions of this study in the establishment of guidelines for the follow-up of pregnant women in the routine obstetric consultations, namely for the planning and implementation of smoking cessation interventions, these findings should be interpreted with caution considering the limitations that can be pointed out to this study. Firstly, this study is exploratory in nature and its transversal design does not allow to establish the direction of causal relations that could be important to understanding the mechanisms involved in the harmful effects of tobacco smoking during pregnancy on women’s health and children’s health and development over time; as well as to assess and discuss the cost-benefits of the implementation of smoking cessation interventions in the routine prenatal care.

Moreover, the different sizes of the comparison groups involved in the analysis of the risk factors associated with smoking during pregnancy have limited the number of comparisons that could be made, as well as the scope of the study. The establishment of the grouping categories was carried out to guarantee the equilibrium of the groups’ size and enable the statistical analyses, which could not be representative of all pregnant women. So, understanding of the determinants of smoking during pregnancy remains inadequate and should still focus attention on future studies.

Another limitation was the lack of an objective and standardized assessment of tobacco use [69], i.e., the categorization of a smoker or non-smoker was only based on the participants’ self-report. For this study, the woman was considered a smoker or non-smoker through her self-report, which may have resulted in an underestimation of the prevalence of tobacco smoking. Even though in 90% of cases this self-report seems to correspond to the objective assessment [70,71]. It is suggested that in future studies a standardized assessment should be performed, to classify the participant as a smoker or non-smoker. Additionally, the reduction in women’s smoking between the period before getting pregnant and during pregnancy was not deeply understood. Despite being asked about their smoking habits during these periods, the categories of answers provided did not allow for a precise analysis of this change. In this sense, it would be important to take this aspect into consideration for future research. Despite the limitations of this study, it was possible to draw relevant conclusions that added knowledge to the research on tobacco consumption during the transition to parenthood, namely on its prevalence, risk factors, health care monitoring, and treatment for smoking cessation in pregnant women.

## 5. Conclusions

Summing up the main conclusions, the present findings indicated that focus on smoking cessation is important in antenatal care as many women smoke before pregnancy and continue to do so in pregnancy, despite the existence of information on the harmful effects of smoking during pregnancy. Tobacco consumption is observed mainly in pregnant women who are single or divorced, lower educated, with clinically relevant anxiety symptoms, and greater exposure to environmental tobacco smoke, either inside or outside the household). This knowledge could be used to tailor preconception prevention strategies as well as interventions during pregnancy, specially designed for high-risk women, with greater benefits for them. Additionally, although most pregnant women are asked about their smoking habits in their regular obstetric consultations, the referral to specialized smoking cessation interventions during pregnancy is almost non-existing, alerting to the urgent need to step up the training of healthcare professionals, both in screening and intervention in tobacco consumption. Psychoeducation with pregnant women, mainly with the ones who smoke can also be an important tool to alert for the harmful impact of smoking on a mother’s health, children’s development, and family well-being, as well as for the intergenerational benefits of integrating a smoking cessation intervention during pregnancy. In fact, the importance of a smoking cessation intervention was more emphasized by women who do not smoke, compared to women who do smoke, precisely the ones who needed and who are at higher risk of developing health problems, suggesting the need to focus on professional attention for these women. Among the pregnant women who perceived the usefulness of smoking cessation interventions during pregnancy, the types of treatments perceived as the most useful and effective included the monitoring and intervention implemented by health care professionals on a daily or weekly basis.

## Figures and Tables

**Figure 1 ijerph-19-06595-f001:**
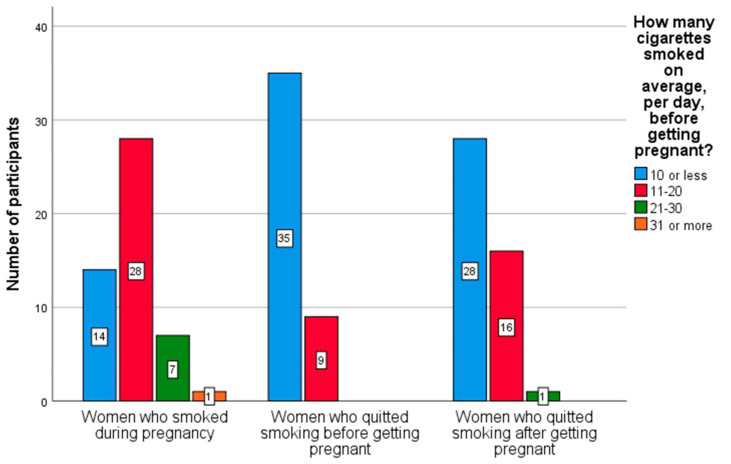
Smoking habits of women before getting pregnant.

**Figure 2 ijerph-19-06595-f002:**
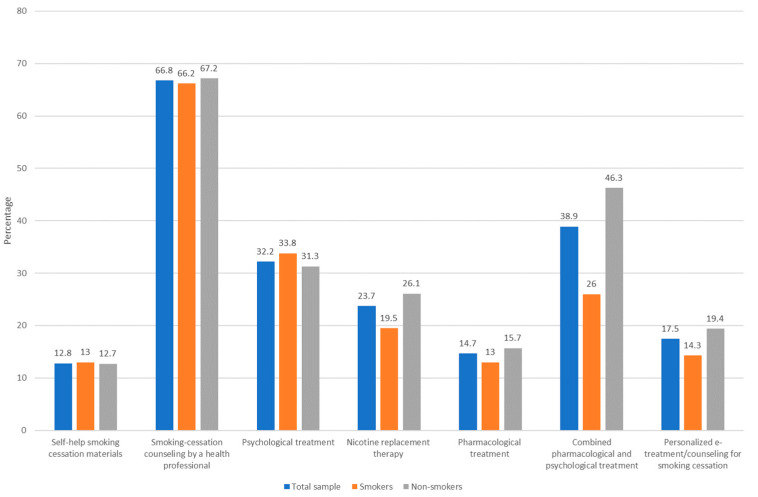
Types of smoking cessation interventions perceived by pregnant women as the most useful and effective during pregnancy (% of affirmative answers).

**Table 1 ijerph-19-06595-t001:** Sociodemographic, obstetric, and psychological characteristics of pregnant women (*N* = 247).

Variables	*N*	%	Mean	Standard-Deviation
Age			30.3	5.02
18–35	204	82.6		
≥36	43	17.4		
Area of residence				
Madeira Archipelago	8	3.2		
Azores Archipelago	5	2.1		
Northern Portugal	97	39.3		
Central Portugal	109	44.1		
Southern Portugal	28	11.3		
Marital status				
Married	130	52.6		
Separated/divorced	4	1.6		
Single	113	45.7		
Education				
Higher education	129	52.2		
High School	107	43.3		
No education/Elementary School	11	4.5		
Occupational status				
Unemployed	44	17.8		
Employed	76	30.8		
Employed but not working (e.g., sick/medical leave)	127	51.4		
Gravidity (number of pregnancies including the current pregnancy)			1.56	0.95
Primigravid	147	59.5		
Multigravid	100	40.5		
Risk pregnancy				
No	157	63.6		
Yes	90	36.4		
Gestational period			24.34	9.38
1st Trimester (1–13 weeks)	37	15		
2nd Trimester (14–26 weeks)	101	40.9		
3rd Trimester (27–40 weeks)	109	44.1		
Diagnosis of a psychological disorder				
Anxiety	7	36.8		
Panic	1	5.3		
Depression	6	31.6		
Chronic depression	1	5.3		
Depression and anxiety	2	10.5		
Obsessive–compulsive disorder	1	5.3		
Post-traumatic stress	1	5.3		
Anxiety			42.62	11.27
High anxiety	145	58.7		
Low anxiety	102	41.3		

**Table 2 ijerph-19-06595-t002:** Associations between the smoking habits of pregnant women and sociodemographic, obstetric, and psychological and environmental tobacco smoke exposure dimensions: chi-squared tests.

Variables	Smokers *n* = 97 %	Non-Smokers *n* = 150 %	Chi-Squared Statistics
Age			χ^2^ (1) = 0.15, *p* = 0.70
18–35	81.4	83.3	
≥36	18.6	16.7	
Marital status			χ^2^ (1) = 4.42, *p* < 0.05
Single + Separated/divorced	55.7	42	
Married	44.3	58	
Education			χ^2^ (1) = 3.02, *p* < 0.10
Higher education	45.4	56.7	
No education/Elementary School + High School	54.6	43.3	
Occupational status			χ^2^ (1) = 0.37, *p* = 0.54
Employed	33	29.3	
Unemployed + Employed but not working	67	70.7	
Gravidity (number of pregnancies including the current pregnancy)			χ^2^ (1) = 0.21, *p* = 0.65
Primigravid	57.7	60.7	
Multigravid	42.3	39.3	
Risk pregnancy			χ^2^ (1) = 2.09, *p* = 0.15
No	69.1	60	
Yes	30.9	40	
Anxiety			χ^2^ (1) = 4.54, *p* < 0.05
Low anxiety	33	46.7	
High anxiety	67	53.3	
Environmental tobacco smoke exposure			χ^2^ (3) = 13.28, *p* < 0.01
Almost every day	27.8	16	
Weekly	18.6	10.7	
Monthly	8.2	4.7	
Less than once a month or never	45.4	68.7	

## Data Availability

Data supporting reported results can be requested to the authors, via email to anac@upt.pt.

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
