# Peer review of "Tobacco Smoking during Pregnancy: Women’s Perception about the Usefulness of Smoking Cessation Interventions"

_ijerph, 2022, doi:10.3390/ijerph19116595_

Round 1

Reviewer 1 Report

This is an interesting topic. The research on tobacco consumption and smoking-cessation programmes, and the implications for public health, including pregnant women and their unborn babies, is a critical issue in recent environmental and public health studies. This manuscript reports on a sample of Portuguese pregnant women. However, a few issues need to be addressed (listed below) prior to publication. 

Abstract

According to the method section, the sample included women aged between 18- and 43-year-old, not ‘ages between 18–35-year-old’ (line 19). Please clarify.

Ethical considerations:

Was ethical approval obtained before the data collection? When/how? From whom/institutions? Please provide further details as the participants of this study are clustered as ‘vulnerable individuals’. (The explanation reported in lines 158-160 does not reveal the above-requested details).

Results

It is recommended to present the summary of Chi-square tests in a table, enabling the reader to find out a specific value, including not statistically significant results.

Consistency in reporting p-value: line 296, p-value should read ‘p = <.001’, please check it. (all p-values in the manuscript are reported lower than 0.05 or 0.01 with this exception in line 296)    

Discussion of the results

As mentioned in the methodology section, the data was collected during the first Covid-related pandemic lockdown (from May to July 2020). It would be very interesting to reflect on whether/how Covid-related restrictions have influenced participants’ smoking behaviours or perceptions regarding smoking-cessation interventions.

Minor issues

There are several typos in the manuscript - see the examples below. The manuscript needs to be checked.

Page 3, line 115: ‘pacient’ should read ‘patient’

Page 3, line 130: ‘pre-pregancy’ should read ‘pre-pregnancy’

Author Response

This is an interesting topic. The research on tobacco consumption and smoking-cessation programmes, and the implications for public health, including pregnant women and their unborn babies, is a critical issue in recent environmental and public health studies. This manuscript reports on a sample of Portuguese pregnant women. However, a few issues need to be addressed (listed below) prior to publication. 

We really like to thank your very kind words and the careful revision of the manuscript. We consider that it allowed a great improvement of the paper.

Abstract

According to the method section, the sample included women aged between 18- and 43-year-old, not ‘ages between 18–35-year-old’ (line 19). Please clarify.

The information included in the abstract was wrong. It was a typo. I'm really sorry about that. This mistake was corrected at the abstract. 

Ethical considerations:

Was ethical approval obtained before the data collection? When/how? From whom/institutions? Please provide further details as the participants of this study are clustered as ‘vulnerable individuals’. (The explanation reported in lines 158-160 does not reveal the above-requested details).

The requested information was added in the method section. Please, find all the details at page 5, line 182-184.

Results

Taking into account the valuable comments of the reviewers, which we would like to thank, we have revised all the results section, adding some additional information associated with the aims of the study, that allow to better understand the results. Moreover, we decided to only include in the paper data from the participants who have completed all the sections of the survey. We consider that as the different sections of the results are interrelated, this will clarify the information presented. This change in the sample size doesn't have implyed any change in the direction of the previously presented results.

It is recommended to present the summary of Chi-square tests in a table, enabling the reader to find out a specific value, including not statistically significant results.

This information was added in the manuscript (please, find it at the Table 2).

Consistency in reporting p-value: line 296, p-value should read ‘p = <.001’, please check it. (all p-values in the manuscript are reported lower than 0.05 or 0.01 with this exception in line 296)    

Thank you very much for your careful revision. We have uniformized the statistics' report.

Discussion of the results

As mentioned in the methodology section, the data was collected during the first Covid-related pandemic lockdown (from May to July 2020). It would be very interesting to reflect on whether/how Covid-related restrictions have influenced participants’ smoking behaviours or perceptions regarding smoking-cessation interventions.

Thank you very much for your valuable suggestion. This aspect was discussed at the Discussion section. Please, find the details at page 13, lines 475-486.

Minor issues

There are several typos in the manuscript - see the examples below. The manuscript needs to be checked.

Page 3, line 115: ‘pacient’ should read ‘patient’

Page 3, line 130: ‘pre-pregancy’ should read ‘pre-pregnancy’

Thank you for your careful revision. The manuscript was checked and typos corrected. Moreover, an extensive English revision was implemented by a specialized reviewer to make the manuscript more clear and understandable.

Reviewer 2 Report

This manuscript covers an important topic, but many statements in the introductory, discussion and conclusion sections are broad in nature. It is, therefore, often difficult to determine if the data reported is from Portugal or another country. Providing more geographically contextual sentences, especially when reporting data from previous studies, will be useful so the reader can recognize how much research already exists in the country vs. outside. In addition, there are many grammatical errors and punctuation issues throughout the manuscript. I have mentioned a few of these below. The authors should consider using a professional editing or proofreading service to improve their work.

Abstract

  • Line 13: I believe the authors meant to write extreme importance, not extremely importance
  • Line 18: Instead of the current phrase, it may be better to say associated with tobacco consumption…
  • Line 19: Consider rephrasing to say pregnant Portuguese women
  • Lines 18-21: Consider rephrasing the sentence, as it is long and convoluted
  • Line 22: I believe the authors meant to write women who smoked, not women who smoke

Introduction

  • Page 2, line 62: The authors should consider replacing the semicolon with a period to end the sentence
  • Page 2, line 65: Can the authors repeat the reference at the end of the sentence so that it is clear they are still describing findings from the study by Alves et al.?
  • Page 2, lines 81-84: Sentence is unclear
  • Can the authors provide more data on the current prevalence of smoking/tobacco consumption and access to tobacco products in Portugal/among the Portuguese? Are there other studies other than the one conducted by Alves et al.? Also, why the focus on tobacco consumption in this population? Is this habit more prevalent than alcohol use or other substance use among pregnant women in Portugal, for example?
  • Page 3, line 106: The authors indicated that health professionals do not always properly address smoking cessation. Is this specifically the case in Portugal or elsewhere?

Materials and Methods

  • Can we assume that study participants were Portuguese women living in Portugal or was geographical location not a consideration for this study? The authors should include a brief statement to indicate where these women lived
  • Page 3, lines 144-145: Instead of saying “the high majority”, can the authors include or provide a percentage or the number of women who did not have a diagnosis? Also, the authors should consider saying pregnant women did not, instead of the current phrase
  • The data in Table 1 indicates that most study participants (51.7%) were on maternity leave while they were still pregnant. Can the authors confirm if that is indeed the case? This information seems a bit unusual as women who work often start maternity leave towards the end of the third trimester/after childbirth unless there are extenuating circumstances. Perhaps, this finding may be unique to Portugal?
  • If participants self-reported whether they had a high-risk pregnancy, can the authors share the definition of high-risk pregnancy that was included in the survey, if any?
  • Was the entire survey available only in Portuguese?
  • Did study participants receive an incentive for completing the survey?
  • Page 6, line 217: I believe it is incorrect to say The authors should consider rephrasing
  • Can the authors include a statement indicating which regulatory body/authority provided ethical approval for the study?

Results

  • In addition to the percentages, can the authors also share the number of pregnant women in each category for smoking habits? Providing these details in a table or some other graphic format would be useful since this is a key aspect of the study
  • Page 7, line 265: It seems a key number is missing from the sentence. How many women started tobacco consumption between the ages of 10 and 26? What do the mean and standard deviation numbers in line 267 represent?
  • Is there stigma around women using tobacco or smoking in Portugal? This may account for why 30 women denied tobacco use even though they are smokers (Page 7, lines 303-305). The authors may want to consider including some information on smoking-related stigma in Portugal within the discussion of findings, if available and as relevant
  • Concerning the categories listed in Figure 1, were these exact categories provided in the survey so that participants could select the intervention they considered the most effective? Was there an open-ended question about perceptions of intervention effectiveness that the authors then collapsed into the listed categories? Providing clarity on this process within the methods section would be valuable.
  • For Figure 1, is there a reason why the label for “pharmacological” is much lower than the other labels on the x axis?
  • Can the authors refer to Figure 2 within the text of the results section?
  • For each figure, it would be useful to include the number of participants as part of the label/heading

Discussion

  • The authors stated certain hypotheses at the end of the introduction section. Early within the discussion section, it would be helpful if they can explicitly state whether the hypotheses were supported or not as well as the related reasons why
  • Page 9, lines 362-363: The authors mentioned that several studies have looked at links between smokers and anxiety, but they only listed one reference. Can they provide more citations for their statement?
  • The authors indicated that pregnant smokers did not receive referrals for cessation interventions; however, I would caution on the interpretation of this finding. Considering that pregnant smokers in this study were the ones who perceived interventions as less useful, is it possible that pregnant smokers did not accurately recall whether they received referrals or that they ignored the referrals because they were not motivated enough to stop their habits?
  • Page 10, lines 414-420: Consider rephrasing the sentence, as it is long and convoluted
  • Are there any studies that have examined knowledge of smoking/tobacco use or the links between knowledge and behavior among Portuguese women? Findings from such research may be useful to include in the study, particularly around the reasons why there is still a high prevalence of pregnant smokers
  • The authors should also consider environmental and social determinants that may affect smoking among Portuguese women

Author Response

This manuscript covers an important topic, but many statements in the introductory, discussion and conclusion sections are broad in nature. It is, therefore, often difficult to determine if the data reported is from Portugal or another country. Providing more geographically contextual sentences, especially when reporting data from previous studies, will be useful so the reader can recognize how much research already exists in the country vs. outside. In addition, there are many grammatical errors and punctuation issues throughout the manuscript. I have mentioned a few of these below. The authors should consider using a professional editing or proofreading service to improve their work.

We really like to thank reviewer's positive feedback about the importance of the study's aim and for the careful revision of the manuscript. We consider that all the reviewer's comments allowed a great improvement of the paper.

Abstract

  • Line 13: I believe the authors meant to write extreme importance, not extremely importance
  • Line 18: Instead of the current phrase, it may be better to say associated with tobacco consumption…
  • Line 19: Consider rephrasing to say pregnant Portuguese women
  • Lines 18-21: Consider rephrasing the sentence, as it is long and convoluted
  • Line 22: I believe the authors meant to write women who smoked, not women who smoke

Thank you for your careful revision. The manuscript was checked and typos corrected. The long and convoluted sentences were also revised to make them more clear. Moreover, an extensive English revision was implemented by a specialized reviewer to make the manuscript more clear and understandable.

Introduction

  • Page 2, line 62: The authors should consider replacing the semicolon with a period to end the sentence

The request was implemented.

  • Page 2, line 65: Can the authors repeat the reference at the end of the sentence so that it is clear they are still describing findings from the study by Alves et al.?

The request was implemented.

  • Page 2, lines 81-84: Sentence is unclear

The sentence was clarified.

  • Can the authors provide more data on the current prevalence of smoking/tobacco consumption and access to tobacco products in Portugal/among the Portuguese? Are there other studies other than the one conducted by Alves et al.? Also, why the focus on tobacco consumption in this population? Is this habit more prevalent than alcohol use or other substance use among pregnant women in Portugal, for example?

New data was added to the Introduction section. Please, find the details at page 1, lines 35-47; page 2, lines 52-56, 63-65 and 76-82.

  • Page 3, line 106: The authors indicated that health professionals do not always properly address smoking cessation. Is this specifically the case in Portugal or elsewhere?

This information is not specific to Portugal. As far as we know, no available information exists for our country. Changes were implemented concerning this aspect at page 3, lines 125-134.

Materials and Methods

  • Can we assume that study participants were Portuguese women living in Portugal or was geographical location not a consideration for this study? The authors should include a brief statement to indicate where these women lived

All the participants are Portuguese living in Portugal. This information was clarified at the manuscript at page 4 (Participants section and Table 2).

  • Page 3, lines 144-145: Instead of saying “the high majority”, can the authors include or provide a percentage or the number of women who did not have a diagnosis? Also, the authors should consider saying pregnant women did not, instead of the current phrase

The requests were implemented. Moreover, based in the valuable comments of the reviewers, which we would like to thank, we have revised all the results section, adding some additional information associated with the aims of the study, that allow to better understand the results. Moreover, we decided to only include in the paper data from the participants who have completed all the sections of the survey. We consider that as the different sections of the results are interrelated, this will clarify the information presented. This change in the sample size doesn't have implyed any change in the direction of the previously presented results.

  • The data in Table 1 indicates that most study participants (51.7%) were on maternity leave while they were still pregnant. Can the authors confirm if that is indeed the case? This information seems a bit unusual as women who work often start maternity leave towards the end of the third trimester/after childbirth unless there are extenuating circumstances. Perhaps, this finding may be unique to Portugal?

Thank you very much for your careful revision. It was an English language mistake. What we really wanted to say is that some of the women were employed but nor working because they were in sick or medical leave. We are really sorry about that mistake. It was corrected. 

  • If participants self-reported whether they had a high-risk pregnancy, can the authors share the definition of high-risk pregnancy that was included in the survey, if any?
  • Was the entire survey available only in Portuguese?
  • Did study participants receive an incentive for completing the survey?

Clarifications were made at page 4, line 174; page 5, line 189; and page 6, line 195, respectively.

  • Page 6, line 217: I believe it is incorrect to say The authors should consider rephrasing

The request was implemented.

  • Can the authors include a statement indicating which regulatory body/authority provided ethical approval for the study?

The requested information was added in the method section. Please, find all the details at page 5, line 182-184.

Results

Taking into account the valuable comments of the reviewers, which we would like to thank, we have revised all the results section, adding some additional information associated with the aims of the study, that allow to better understand the results. Moreover, we decided to only include in the paper data from the participants who have completed all the sections of the survey. We consider that as the different sections of the results are interrelated, this will clarify the information presented. This change in the sample size doesn't have implyed any change in the direction of the previously presented results. So, all the reviewer's requests were implemented taking into account the new sample considered and respective statistical analyses implemented.

  • In addition to the percentages, can the authors also share the number of pregnant women in each category for smoking habits? Providing these details in a table or some other graphic format would be useful since this is a key aspect of the study
  • Page 7, line 265: It seems a key number is missing from the sentence. How many women started tobacco consumption between the ages of 10 and 26? What do the mean and standard deviation numbers in line 267 represent?
  • Is there stigma around women using tobacco or smoking in Portugal? This may account for why 30 women denied tobacco use even though they are smokers (Page 7, lines 303-305). The authors may want to consider including some information on smoking-related stigma in Portugal within the discussion of findings, if available and as relevant

Thank you very much for your careful revision. It was a mistake due to a difficulty in English language expression. The sentence was  rephrased. 

  • Concerning the categories listed in Figure 1, were these exact categories provided in the survey so that participants could select the intervention they considered the most effective? Was there an open-ended question about perceptions of intervention effectiveness that the authors then collapsed into the listed categories? Providing clarity on this process within the methods section would be valuable.

This information was clarified, namely when the instruments section (pages 6-7, lines 203-281).

  • For Figure 1, is there a reason why the label for “pharmacological” is much lower than the other labels on the x axis?
  • Can the authors refer to Figure 2 within the text of the results section?
  • For each figure, it would be useful to include the number of participants as part of the label/heading

All these comments were taken into consideration when the Results section was revised.

Discussion

  • The authors stated certain hypotheses at the end of the introduction section. Early within the discussion section, it would be helpful if they can explicitly state whether the hypotheses were supported or not as well as the related reasons why

The request was taken into consideration. For details, please see page 12, lines 444-468.

  • Page 9, lines 362-363: The authors mentioned that several studies have looked at links between smokers and anxiety, but they only listed one reference. Can they provide more citations for their statement?

A new reference was added. A more deeply discussion about the association between anxiety and smoking was also implemented. For details, please see page 13, lines 495-511.

  • The authors indicated that pregnant smokers did not receive referrals for cessation interventions; however, I would caution on the interpretation of this finding. Considering that pregnant smokers in this study were the ones who perceived interventions as less useful, is it possible that pregnant smokers did not accurately recall whether they received referrals or that they ignored the referrals because they were not motivated enough to stop their habits?

Thank you very much for your comments. A more deeply discussion about this issue was implemented. Please, see it at pages 13-14, lines 520-539.

  • Page 10, lines 414-420: Consider rephrasing the sentence, as it is long and convoluted

The request was implemented.

  • Are there any studies that have examined knowledge of smoking/tobacco use or the links between knowledge and behavior among Portuguese women? Findings from such research may be useful to include in the study, particularly around the reasons why there is still a high prevalence of pregnant smokers
  • The authors should also consider environmental and social determinants that may affect smoking among Portuguese women

Thank you very much for your comments. A more deeply discussion around these issues was implemented. Please, see it at page 14, lines 540-579.

Reviewer 3 Report

This is a well-intentioned paper on an important topic, and the authors deserve credit for both their interest in the issue and the aims of their research. Unfortunately, there is a major obstacle for a reviewer, which reflects less on the author and the paper than on the use of English as the language for most major journals. I should stress that I sympathise with the authors but as a reviewer, I can only provide comments on the paper as it has been submitted and provided to me. The paper appears to have been translated into English; but there are so many language and style errors and other problems that it is not possible to provide a proper review. These apply to language and style throughout the paper, and there are so many that it would not be possible to list them all. Unfortunately, many of them make the paper hard to understand,  that the paper would benefit from editing/translation by someone with a stronger grasp of English.

Author Response

This is a well-intentioned paper on an important topic, and the authors deserve credit for both their interest in the issue and the aims of their research. Unfortunately, there is a major obstacle for a reviewer, which reflects less on the author and the paper than on the use of English as the language for most major journals. I should stress that I sympathise with the authors but as a reviewer, I can only provide comments on the paper as it has been submitted and provided to me. The paper appears to have been translated into English; but there are so many language and style errors and other problems that it is not possible to provide a proper review. These apply to language and style throughout the paper, and there are so many that it would not be possible to list them all. Unfortunately, many of them make the paper hard to understand,  that the paper would benefit from editing/translation by someone with a stronger grasp of English.

Thank you very much for your positive feedback about the importance of the topic. We are really sorry that the quality of the English language and style didn't have allowed the reviewer to provide comments on the paper. The manuscript was checked and typos corrected. Moreover, an extensive English revision was implemented by a specialized reviewer to make the manuscript more clear and understandable.

Round 2

Reviewer 2 Report

The manuscript on tobacco smoking among women in Portugal has promise, but it still needs some more work. In many areas, the writing can be simplified and improved to ensure clarity. For example, there are still many run-on sentences throughout the manuscript but more so in the discussion section. The conclusion could also be a bit tighter if one or two main takeaways were more clearly stated. I have included some more detailed notes, which I hope may be useful for the authors.

Introduction

  • Page 1, lines 41-42: Consider reframing the statement and include some punctuation.
  • Pages 1-2, lines 46-47: It seems awkward to say that women’s smoking habits faced a specific challenge. Consider revising the statement
  • Page 2, lines 54-56: Consider deleting substances at the end of the statement as it is redundant
  • Page 2, lines 79-82: The first statement here is incomplete, and the second statement is unclear. Can the authors briefly state the benefit of identifying the risk factors?
  • Page 3, lines 143-153: The aims of the study provided here are somewhat difficult to follow, especially since a lot of the text is in parenthesis. The authors should consider breaking the aims into three succinct sentences

Methods

  • Page 7, lines 264-265: It is unclear if this is supposed to be a heading or something else. The statement is incomplete
  • Page 7, lines 266-269: The authors mentioned that they used a “questionnaire specifically built for the current study” to collect data. Did they not do the same thing for questions on smoking and environmental tobacco smoke exposure, which are mentioned earlier on the same page? The description provided creates the impression that there were several questionnaires being used in this study, as opposed to one questionnaire with various sections
  • Page 7, lines 269-277: This statement is long and has several details. The authors should consider breaking it up into multiple statements
  • The analyses section is a bit confusing. The authors should consider reorganizing this section so that type of analyses that they did for each aim becomes more apparent.

Results

  • The labels and heading for Figure 1 are confusing. For example, what does it mean if someone “stopped smoking after to get pregnant”? Also, if some women stopped smoking before getting pregnant, does not it not seem strange to include them on an axis labeled “smoking habits during pregnancy”?
  • Page 9, line 351: What does it mean if “40 referred to smoke 10 or less cigarettes”? If the interpretation is 40 women indicated that they smoked 10 or less cigarettes, then the authors need to specifically state that and also check the use of the phrase “referred” throughout the results section
  • The label for Figure 2 is a bit deceptive. Pregnant women who have never smoked may have indicated the smoking-cessation intervention that they perceived to be the most useful. However, that doesn’t mean the intervention was effective for them because it would never had applied to them personally.
  • It looks like the numbers on top of each bar for Figure 2 should have a period for the decimals and not a comma. For example, it should say 67.2 (not 67,2) for non-smokers who considered smoking-cessation counseling to be the most useful intervention

Author Response

The manuscript on tobacco smoking among women in Portugal has promise, but it still needs some more work. In many areas, the writing can be simplified and improved to ensure clarity. For example, there are still many run-on sentences throughout the manuscript but more so in the discussion section.

We are really grateful for your revision of the revised manuscript. Your additional comments were all considered and contributed to the improvement of the paper. The revisions made to the manuscript were marked up using the “Track Changes” function.

The conclusion could also be a bit tighter if one or two main takeaways were more clearly stated. I have included some more detailed notes, which I hope may be useful for the authors.

The Discussion and Conclusion sections were revised in order to insure higher clarity. The conclusion was strenghted by the inclusion of more concrete implications of the findings. The revisions made were marked up using the “Track Changes” function.

Introduction

  • Page 1, lines 41-42: Consider reframing the statement and include some punctuation.

The statement was revised. The revisions made were marked up using the “Track Changes” function. 

  • Pages 1-2, lines 46-47: It seems awkward to say that women’s smoking habits faced a specific challenge. Consider revising the statement

The statement was revised. The revisions made were marked up using the “Track Changes” function. 

  • Page 2, lines 54-56: Consider deleting substances at the end of the statement as it is redundant

"Substances" word was deleted as suggested.

  • Page 2, lines 79-82: The first statement here is incomplete, and the second statement is unclear. Can the authors briefly state the benefit of identifying the risk factors?

The statements were revised to clarification. The benefits of identifying the risk factors were briefly clarifyed. The revisions made were marked up using the “Track Changes” function. 

  • Page 3, lines 143-153: The aims of the study provided here are somewhat difficult to follow, especially since a lot of the text is in parenthesis. The authors should consider breaking the aims into three succinct sentences

The paragraph was revised in order to make easier to identify the main aims. The revisions made were marked up using the “Track Changes” function.

Methods

  • Page 7, lines 264-265: It is unclear if this is supposed to be a heading or something else. The statement is incomplete
  • Page 7, lines 266-269: The authors mentioned that they used a “questionnaire specifically built for the current study” to collect data. Did they not do the same thing for questions on smoking and environmental tobacco smoke exposure, which are mentioned earlier on the same page? The description provided creates the impression that there were several questionnaires being used in this study, as opposed to one questionnaire with various sections

The reviewer is perfectly right. A single questionnaire with various sections was used to assess smoking habits, tobacco smoke exposure, and women's perceptions about the usefulness of smoking-cessation interventions. The Instruments section was revised in order to clarify both comment.  The revisions made were marked up using the “Track Changes” function.

  • Page 7, lines 269-277: This statement is long and has several details. The authors should consider breaking it up into multiple statements

The statement was revised in order to make it easy to follow. The revisions made were marked up using the “Track Changes” function.

  • The analyses section is a bit confusing. The authors should consider reorganizing this section so that type of analyses that they did for each aim becomes more apparent.

The comment was taken into consideration and the analyses section changed in accordance. The revisions made were marked up using the “Track Changes” function.

Results

  • The labels and heading for Figure 1 are confusing. For example, what does it mean if someone “stopped smoking after to get pregnant”? Also, if some women stopped smoking before getting pregnant, does not it not seem strange to include them on an axis labeled “smoking habits during pregnancy”?

The reviewer is perfectly right. Clarifications were made and changes were implemented in accordance. The revisions made were marked up using the “Track Changes” function.

  • Page 9, line 351: What does it mean if “40 referred to smoke 10 or less cigarettes”? If the interpretation is 40 women indicated that they smoked 10 or less cigarettes, then the authors need to specifically state that and also check the use of the phrase “referred” throughout the results section

Clarifications were implemented.  The revisions made were marked up using the “Track Changes” function.

  • The label for Figure 2 is a bit deceptive. Pregnant women who have never smoked may have indicated the smoking-cessation intervention that they perceived to be the most useful. However, that doesn’t mean the intervention was effective for them because it would never had applied to them personally.

We are grateful for the comment, which was taken into consideration. The revisions made were marked up using the “Track Changes” function.

  • It looks like the numbers on top of each bar for Figure 2 should have a period for the decimals and not a comma. For example, it should say 67.2 (not 67,2) for non-smokers who considered smoking-cessation counseling to be the most useful intervention

A new graphic was included in order to integrate the reviewer's comment.